# Dry reforming of methane over gallium-based supported catalytically active liquid metal solutions

Moritz Wolf [1,2], Ana Luiza de Oliveira[1,2], Nicola Taccardi[1], Sven Maisel[3], Martina Heller[4], Sharmin Khan Antara[1], Alexander Søgaard[1], Peter Felfer[4], Andreas Görling [3], Marco Haumann [1] & Peter Wasserscheid [1,2 ✉]

Gallium-rich supported catalytically active liquid metal solutions (SCALMS) were recently introduced as a new way towards heterogeneous single atom catalysis. SCALMS were demonstrated to exhibit a certain resistance against coking during the dehydrogenation of alkanes using Ga-rich alloys of noble metals. Here, the conceptual catalytic application of SCALMS in dry reforming of methane (DRM) is tested with non-noble metal (Co, Cu, Fe, Ni) atoms in the gallium-rich liquid alloy. This study introduces SCALMS to high-temperature applications and an oxidative reaction environment. Most catalysts were shown to undergo severe oxidation during DRM, while Ga-Ni SCALMS retained a certain level of activity. This observation is explained by a kinetically controlled redox process, namely oxidation to gallium oxide species and re-reduction via $H_2$ activation over Ni. Consequentially, this redox process can be shifted to the metallic side when using increasing concentrations of Ni in Ga, which strongly suppresses coke formation. Density-functional theory (DFT) based ab initio molecular dynamics (AIMD) simulations were performed to confirm the increased availability of Ni at the liquid alloy-gas interface. However, leaching of gallium via the formation of volatile oxidic species during the hypothesised redox cycles was identified indicating a critical instability of Ga-Ni SCALMS for prolonged test durations.

[1] Friedrich-Alexander-Universität Erlangen-Nürnberg (FAU), Lehrstuhl für Chemische Reaktionstechnik (CRT), Egerlandstr. 3, 91058 Erlangen, Germany. [2] Forschungszentrum Jülich, Helmholtz Institute Erlangen-Nürnberg for Renewable Energy (IEK 11), Cauerstr. 1, 91058 Erlangen, Germany. [3] Friedrich-Alexander-Universität Erlangen-Nürnberg (FAU), Lehrstuhl für Theoretische Chemie, Egerlandstr. 3, 91058 Erlangen, Germany. [4] Friedrich-Alexander-Universität Erlangen-Nürnberg (FAU), Lehrstuhl für Werkstoffwissenschaften (Allgemeine Werkstoffeigenschaften), Martensstr. 5, 91058 Erlangen, Germany. ✉email: peter.wasserscheid@fau.de

D ry reforming of methane (DRM, Eq. 1) oxidatively converts $CH_4$ with $CO_2$ to synthesis gas, which makes this process a promising alternative for a sustainable valorisation of methane. The integration of $CO_2$ in the production of $H_2$ and CO is ecologically highly attractive and a first step towards a circular economy[1–5]. The strong C–H bonds in $CH_4$ in combination with the mildly oxidising $CO_2$ make DRM a highly endothermic process. The operation temperatures for DRM are in the range of 700–1000 °C[3], which induces rapid catalyst deactivation via coking, oxidation of the active metallic phase, and sintering[4]. In fact, long-term stability is one of the major challenges in the commercialisation of DRM. Hence, recent advances in catalyst research for DRM mostly focus on concepts to prevent excessive sintering and carbon deposition to improve the stability and long-term performance[5–9].

$$CH_{4(g)} + CO_{2(g)} \rightarrow 2H_{2(g)} + 2CO_{(g)} \qquad (1)$$

Sintering of the active phase and coke formation are classical deactivation pathways for heterogeneous catalysts[10]. Sintering of Ni-based catalysts during DRM is often discussed in the literature[5,11–16]. Several recent studies focus on novel reactor and catalyst concepts to suppress (excessive) carbon deposition in order to enable long-term operability[5,17–19]. Two innovative approaches employing liquid alloys as the catalytically active phase in a bubble column reactor demonstrated the successful elimination of both deactivation mechanisms for DRM[18]. However, the approach with liquid alloys in a bubble column reactor suffers of practical issues, because of the highly corrosive nature of the liquid metals and large required quantities resulting in high reactor costs.

Such limitations in the applicability may be overcome by combining supported liquid phase (SLP) catalysis with liquid alloys. In general, SLP catalysis merges advantages from classical heterogeneous and homogeneous catalysis, such as easy handling of the catalyst and well-defined active sites, respectively[20]. However, the application of SLP catalysts with supported organic liquids, ionic liquids, or molten salts is limited due to the restricted thermal stability of these liquid phases[21–24]. Recently, we introduced a new class of SLP catalysts, namely gallium-rich supported catalytically active liquid metal solutions (SCALMS), which may extend the application of SLP catalysis to high-temperature processes due to the low metal vapour pressures (boiling point of Ga: 2400 °C) and high thermal stability of liquid metals[20,25]. Application of SCALMS materials in the dehydrogenation of propane resulted in suppression of carbon deposition[26–29], which enables extended dehydrogenation cycle times in potential commercial applications[29,30]. SCALMS materials are composed of supported droplets of a liquid alloy consisting of a catalytically active metal and an excess of a low melting metal[20,25,26,28,29]. The insolubility of reactants and products in liquid metals confines the catalytic reaction over SCALMS to the liquid metal/gas interface[20,26,31–35]. The active sites in SCALMS have been described as single atoms of the secondary metal in a Ga matrix[20,26,33–35].

Herein, we report the first application of SCALMS in a high-temperature process exceeding 700 °C. The performance of Ga-rich SCALMS using mesoporous SiC as carrier material was conceptually evaluated during dry reforming of methane (DRM) in the temperature range of 700–1000 °C[3]. Furthermore, DRM over SCALMS also represents the first application of this SLP catalysis concept for oxidative reaction mixtures. This is an essential difference due to the high oxophilicity of Ga. In our study, the evaluation of the catalytic performance of Ga-rich SCALMS with secondary metals that are potential candidates for catalysing DRM (Co, Cu, Fe, Ni) is accompanied by microscopic characterisation of the materials prepared. Moreover, the SCALMS materials are thoroughly analysed after their catalytic application to understand the mechanisms at play.

## Results and discussion

**Catalyst characterisation.** Four abundant metals, namely Co, Cu, Fe and Ni, were selected as potential candidates for SCALMS dry reforming of methane (DRM) catalysts. Gallium-based SCALMS were prepared with these secondary metals via impregnation of pre-synthesised gallium-decorated mesoporous β-SiC (BET surface area of 23 $m^2\,g^{-1}$; Figs. S1–S3). Analysis of the metal content resulted in Ga loadings of 4.5-4.9 wt.% and Ga/metal ratios close to the targeted value of 50 (Table 1). According to the corresponding bimetallic phase diagrams[36–39], all of these Ga-rich alloys are expected to be present in the liquid state at temperatures exceeding 700 °C. Furthermore, passivation of the prepared SCALMS by a thin layer of $Ga_xO$ species has been previously reported but may be easily reduced during catalysis by in situ formed reductants, such as $H_2$[26,28,29]. This observation has been assigned to $H_2$ activation by the secondary metal allowing reduction of $Ga_xO$ species at low temperatures.

The as-prepared SCALMS materials were characterised by means of scanning electron microscopy with elemental mapping via energy-dispersive X-ray spectroscopy (SEM-EDX) to evaluate the morphology of the Ga-rich droplets on the external surface of the β-SiC support material and the distribution of the active metal (Fig. 1; Figs. S4–S5). The Ga-droplet decoration in SCALMS was previously achieved via impregnation followed by the thermal decomposition of gallane complexes $(Et_3N)GaH_3$[20,26,28,29]. When using ultrasound emulsification, the Ga droplets are sonochemically produced from elemental Ga, dispersed in a solvent, and can be directly deposited on the substrate[40]. Even though the rough outer surface of the mesoporous support may enhance physical stabilisation, the droplets showed a certain degree of clustering. A wide size distribution of the droplets was observed, ranging from

**Table 1 Chemical composition of the supported catalytically active liquid metal solutions (SCALMS) prepared.**

| Catalyst | Metal loading (as prepared)/wt.% | | Molar ratio | Metal loading (spent)[a]/wt.% | |
|---|---|---|---|---|---|
| | Ga | Secondary metal | | Ga | Secondary metal |
| $Ga_{47}Co$/SiC | 4.85 | 0.09 | 47.2 | 3.68 | 0.08 |
| $Ga_{52}Cu$/SiC | 4.91 | 0.09 | 51.6 | 4.76 | 0.08 |
| $Ga_{44}Fe$/SiC | 4.53 | 0.08 | 43.7 | 3.46 | 0.08 |
| $Ga_{45}Ni$/SiC | 4.52 | 0.08 | 45.3 | 4.25 | 0.08 |
| $Ga_{12}Ni$/SiC | 5.44 | 0.38 | 12.1 | 4.44 | 0.39 |
| | | | | 3.70 | 0.37 (100 h TOS) |

[a]Loadings of the spent catalysts were corrected for the amount of carbon as identified by means of TGA.

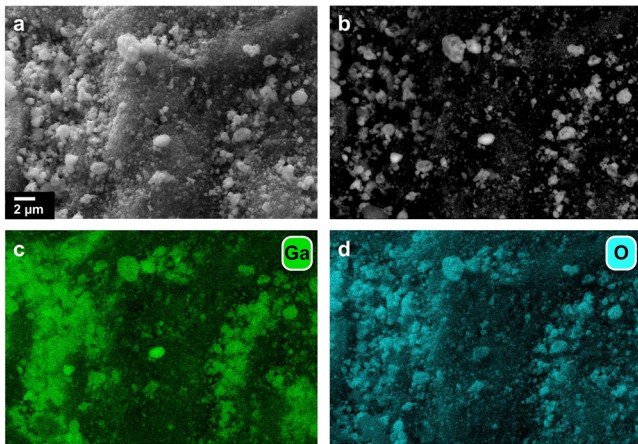

**Fig. 1 Morphology of gallium-rich phase on SiC. a** Scanning electron micrograph, **b** backscattered micrograph and elemental mapping of **c** gallium and **d** oxygen via energy-dispersive X-ray spectroscopy of $Ga_{45}Ni/SiC$ with metal loadings of Ga and Ni of 4.52 and 0.08 wt.%, respectively.

less than 0.1 up to 3 μm. The low concentration of the secondary metals (<0.09 wt.%) resulted in low signal-to-noise ratios for the SEM-EDX elemental maps. Nevertheless, the acquired maps suggest a successful deposition of Ni onto the Ga phase (Fig. S5).

**Catalyst testing.** The performance of the four SCALMS systems during DRM was evaluated at 900 °C and atmospheric pressure in a quartz tube fixed-bed reactor with an equimolar feed of 3:1:1 $Ar:CH_4:CO_2$. The blank quartz tube reactor displayed negligible conversion levels (<1% for $CH_4$ and below detection limit for $CO_2$; Fig. S6). Of the four metals studied, the presence of Ni within the liquid Ga matrix in the $Ga_{45}Ni/SiC$ SCALMS resulted in the highest conversion of $CH_4$ and $CO_2$ (Fig. 2). No increase of activity was observed when comparing the $Ga_{44}Fe/SiC$ SCALMS catalyst with the Ga/SiC benchmark without any secondary metal (Fig. S7). The initial conversion of $CH_4$ and $CO_2$ was the highest for $Ga_{52}Cu/SiC$ SCALMS (Fig. 2). However, this catalyst rapidly deactivated and displayed the activity level of the Ga/SiC reference after 2 h time on stream (TOS). Aside from the Ga-Ni SCALMS, the $Ga_{43}Co/SiC$ SCALMS was the only other system displaying a certain level of DRM activity. However, the conversion levels of this Co-based SCALMS system rapidly decreased from an initial conversion of $CH_4$ of 14% to 7% after 5 h TOS, close to the conversion of the Ga/SiC reference. While the initial conversion of $CH_4$ was only marginally improved over the $Ga_{45}Ni/SiC$ SCALMS, an initial activation period of one hour was observed obtaining conversion levels of $CH_4$ and $CO_2$ of 24 and 34%, respectively (Fig. 2). Subsequently, the Ga-Ni SCALMS slowly deactivated over 20 h TOS to 8 and 11% conversion of $CH_4$ and $CO_2$, respectively.

Interestingly, $CO_2$ conversion in the Ga-Ni and Ga-Co SCALMS catalysed DRM is 50% higher than $CH_4$ conversion throughout the experiment (Fig. 2; Fig. S8). The Ga-Cu and Ga-Fe SCALMS also result in an initially enhanced conversion of $CO_2$ prior to deactivation within the first hours of TOS. This distinct deviation from the equimolar conversion of the reactants suggests an additional conversion pathway for $CO_2$, which ceases in the case of strongly deactivating $Ga_{52}Cu/SiC$ or barely active $Ga_{44}Fe/SiC$ within 5 h TOS (Fig. S8).

**Proposed reaction network.** Activation of $CH_4$ during thermo-catalytic DRM can be described by the formation of $H_2$ alongside adsorbed $CH_x$ or monoatomic carbon (Fig. 3, reaction 1)[41]. In a

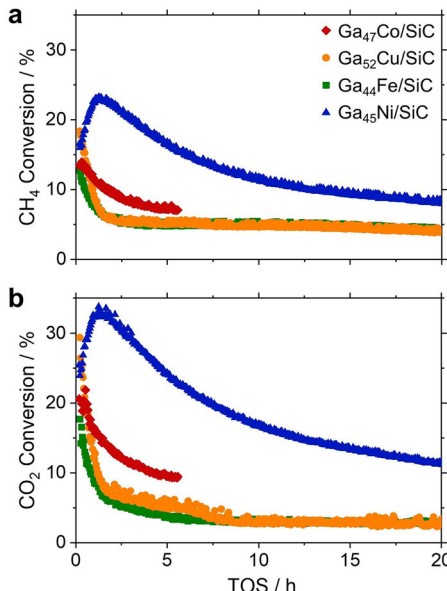

**Fig. 2 Dry reforming of methane over supported catalytically active liquid metal solutions (SCALMS). a** Conversion of $CH_4$ and **b** $CO_2$ using various active metals in gallium-rich alloys employing a mesoporous SiC support. Metal loadings: 4.85 wt.% Ga and 0.09 wt.% Co for $Ga_{47}Co/SiC$, 4.91 wt.% Ga and 0.09 wt.% Cu for $Ga_{52}Cu/SiC$, 4.53 wt.% Ga and 0.08 wt.% Fe for $Ga_{44}Fe/SiC$, 4.52 wt.% Ga and 0.08 wt.% Ni for $Ga_{45}Ni/SiC$. Reaction conditions: 900 °C, 1 bar, 1 g SCALMS, $CH_4:CO_2:Ar = 1:1:3$, 3 $L_N g_{cat}^{-1} h^{-1}$.

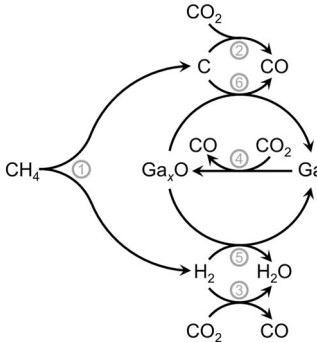

**Fig. 3 Proposed reaction network for dry reforming of methane over gallium-based supported catalytically active liquid metal solutions (SCALMS).** Reaction 1: Activation of methane; reaction 2: Boudouard reaction; reaction 3: reverse water gas shift reaction; reaction 4: oxidation of $Ga^0$ by carbon dioxide; reaction 5: reduction of $Ga_xO$ by in situ formed hydrogen; reaction 6: reduction of $Ga_xO$ by in situ formed monoatomic carbon.

simplified reaction network, monoatomic carbon is then gasified via the Boudouard reaction yielding CO (Fig. 3, reaction 2). The formation of CO is typically increased to the sacrifice of $H_2$ due to the additional conversion of $CO_2$ via the reverse water gas shift (RWGS) reaction (Fig. 3, reaction 3). Hence, synthesis gas from DRM is typically rich in CO and the compositions reported in the literature vary in $H_2/CO$ ratios from 0.8 to equimolarity[42,43]. Herein, the high reaction temperature of 900 °C allows for enhanced conversion of $CO_2$ via the RWGS reaction when compared to lower temperatures (Fig. S9). This was confirmed in dedicated experiments studying the extent of RWGS over Ga-Ni SCALMS and the SiC support material (Fig. S10). While, the bare support material resulted in low conversion levels, almost equilibrium conversion was achieved for $CO_2$ over the $Ga_{45}Ni/SiC$ SCALMS at 900 °C. Hence, the

RWGS reaction is expected to play an important role as a side reaction during DRM and increases the conversion of $CO_2$. However, additional conversion pathways for $CO_2$ are at play as the obtained $H_2/CO$ ratio during DRM over $Ga_{45}Ni/SiC$ stabilises at a value of 0.55 after 1 h TOS (Fig. S8).

Gallium is highly oxophilic with a low standard Gibbs free energy ($\Delta G_{rxn}^0$) of the oxidation reaction with $O_2$ forming $Ga_2O_3$ of -1996.0 kJ mol$^{-1}$ (Fig. S12a), which is more than twice as high as for full oxidation of pyrophoric metallic Co to $Co_3O_4$ (−802.1 kJ mol$^{-1}$). Expectedly, oxidation of liquid $Ga^0$ by mildly oxidative $CO_2$ (Fig. 3, reaction 4) is also feasible with a Gibbs free energy at reaction temperature (900 °C) of −160.7 kJ mol$^{-1}$ (Fig. S12b). In addition, partial oxidation of $Ga^0$ is even more likely than full oxidation to $Ga_2O_3$ in the present system. Nevertheless, bulk oxidation of $Ga^0$ is unlikely as passivation by a thin $Ga_xO$ layer has been previously identified after exposure to the ambient atmosphere during the preparation of SCALMS[26,28,29]. Hence, we hypothesise additional conversion of $CO_2$ to CO during DRM via partial oxidation of metallic liquid Ga (Fig. 3, reaction 4).

The presence of $H_2$ may enable the reduction of surface oxidic gallium species even though the direct reduction of pure $Ga_2O_3$ is thermodynamically restricted at 900 °C. However, the reduction of $Ga_xO$ has recently been monitored during the dehydrogenation of propane over Ga-Rh and Ga-Pt SCALMS at 500 °C by means of in situ high-resolution thermogravimetric analysis coupled with mass spectrometry (HRTGA-MS). This observation was linked to the presence of the secondary metal for the activation of $H_2$ and for the reduction of the gallium-oxygen binding strength[28,29]. Hence, the presence of a dissolved metal at the $Ga_xO$ interface, such as Ni, changing the gallium-oxygen binding energies and being capable of $H_2$ activation can be expected to allow for the reduction of adjacent $Ga_xO$ species (Fig. 3, reaction 5), which has recently been observed under mild conditions (300 °C) by our group for Ga-Ni SCALMS[44]. Reasonably, the mechanism may be similar to $H_2$ spillover from promoters to the active metal in conventional heterogeneous catalysts[45], i.e., $H_2$ is adsorbed and dissociated on the active secondary metal providing adjacent $Ga_xO$ species with activated hydrogen. Hence, the presence of the secondary metal at the interface will also strongly affect the physico-chemical properties, while the partially oxidised $Ga_xO$ species are generally expected to deviate from the thermodynamically stable $Ga_2O_3$. Lastly, the reverse reaction, that is oxidation of liquid $Ga^0$ by $H_2O$, has been reported to be kinetically hindered on some metals due to the high stability of hydroxyl groups[46–49].

Monoatomic carbon from $CH_4$ activation may also serve as a reducing agent for $Ga_xO$ species (Fig. 3, reaction 6). Thermodynamic calculations reveal a strong temperature dependency of the reduction of $Ga_2O_3$ by graphite to $Ga^0$ and CO indicating a small Gibbs free energy at reaction conditions (900 °C) of 56.8 kJ mol$^{-1}$ (Fig. S12b). Nevertheless, reduction via this pathway may become feasible for thermodynamically less stable, highly reactive monoatomic carbon in combination with partially oxidised $Ga_xO$ species (Fig. 3, reaction 6)[50]. Hence, the closed redox cycle with reduction by monoatomic carbon potentially allows for the Boudouard reaction via a Mars-van Krevelen-type mechanism with oxidation of Ga by $CO_2$ filling the oxygen vacancies formed by reduction with carbon. However, the impact of this reaction on the kinetically controlled $Ga^0$-$Ga_xO$ redox process may be limited when compared to the reduction pathway via activated $H_2$ (Fig. 3, reaction 5). Furthermore, only oxidation of $Ga^0$ by $CO_2$ in combination with a preferential reduction of $Ga_xO$ species via $H_2$ activation on the secondary metal (Fig. 3, reaction 3) results in an enhanced conversion of $CO_2$ when compared to $CH_4$.

Similar redox cycles of the active metal phase during DRM have been hypothesised in the literature. Ruckenstein et al. reported a solid solution of CoO in MgO forming small clusters, which hindered sintering during DRM[51]. Further, the strong interaction suppressed excessive coking. The authors hypothesised that the active Co atoms are constantly oxidised by $CO_2$ and $H_2O$ and reduced by $CH_4$ and $H_2$ during DRM at 900 °C, which was key to achieving high stability and catalytic activity for a 12 wt.% Co/MgO catalyst. More recently, McFarland et al. performed combined dry reforming and pyrolysis of $CH_4$ in a molten metal bubble column reactor using a $Ni_{1.86}In$ alloy[18]. Continuous redox cycles were hypothesised for the metallic In phase. In line with early studies by Otsuka et al.[52], cyclic oxidation of In and reduction of $In_2O_3$ allowed for the conversion of $CO_2$ in the molten metal bubble column reactor, while the oxidic intermediate serves as an oxygen shuttle between $CO_2$ and $CH_4$[18].

**Role of nickel concentration**. A $Ga_{12}Ni/SiC$ SCALMS with an increased concentration of Ni (Table 1) was prepared to study the hypothesised promotional effect of active metal atoms on the kinetically controlled $Ga^0$-$Ga_xO$ redox reaction and consequentially the activity and long-term stability during DRM. This alloy composition is expected to have a liquidus line at approx. 600 °C[39]. The increased concentration of Ni in the SCALMS enabled elemental mapping of the active metal using SEM-EDX (Fig. 4; Fig. S13), while the general morphology of the supported Ga phase resembles the as-prepared $Ga_{45}Ni/SiC$ SCALMS (Fig. 1). The analysis confirms co-location of Ni and Ga, which is a prerequisite for the formation of SCALMS upon in situ reduction of the $Ga_xO$ passivation layer and calcined Ni precursors. Similar to the Ga-Ni SCALMS with a lower concentration of Ni, such an activation within the first hour of DRM at 900 °C was also observed for the $Ga_{12}Ni/SiC$ SCALMS (Fig. 5). The maximum $CH_4$ conversion level after this activation period was significantly increased from 24 up to 59%. Subsequently, the Ni-enriched SCALMS also deactivated to a $CH_4$ conversion of 26%. However, contrary to the $Ga_{45}Ni/SiC$ SCALMS that underwent a steady deactivation, the conversion over the $Ga_{12}Ni/SiC$ SCALMS stabilised after 15 h TOS and was demonstrated to be stable over 100 h TOS in a dedicated long-term experiment (Fig. S14). Note, that the specific activity of Ni of approx. 0.1 mol$_{CH4}$ h$^{-1}$ g$_{Ni}^{-1}$ is inferior when compared to classical catalyst

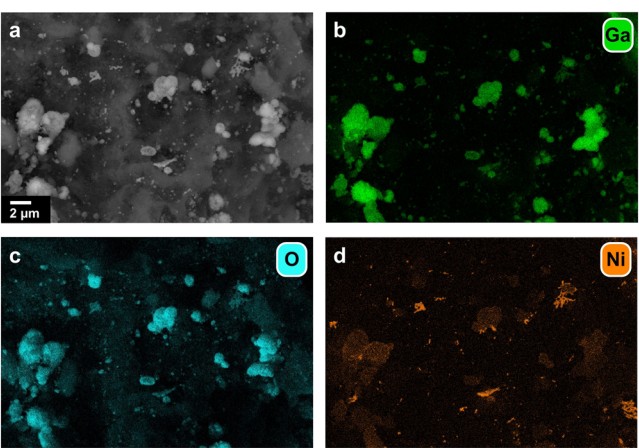

**Fig. 4 Morphology and metal distribution in with higher nickel concentration. a** Backscattered scanning electron micrograph and elemental mapping of **b** gallium, **c** oxygen, and **d** nickel via energy-dispersive X-ray spectroscopy of $Ga_{12}Ni/SiC$ with metal loadings of Ga and Ni of 5.44 and 0.38 wt.%, respectively.

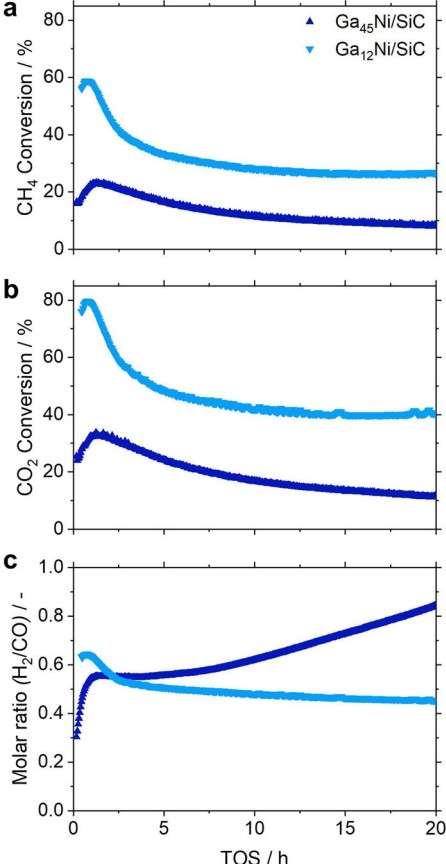

**Fig. 5 Effect of nickel concentration during dry reforming of methane.**
Conversion of **a** $CH_4$ and **b** $CO_2$, as well as **c** the obtained $H_2/CO$ ratio in the product gas during dry reforming of methane over Ga-Ni SCALMS with different concentrations of the active metal Ni employing a mesoporous SiC support. Metal loadings: 4.52 wt.% Ga and 0.08 wt.% Ni for $Ga_{45}Ni/SiC$, 5.44 wt.% Ga and 0.38 wt.% Ni for $Ga_{12}Ni/SiC$. Reaction conditions: 900 °C, 1 bar, 1 g SCALMS, $CH_4:CO_2:Ar = 1:1:3$, 3 $L_N$ $g_{cat}^{-1}$ $h^{-1}$.

concepts, which may reach up to 100 times higher specific conversion rates[53]. The stabilisation at higher conversion levels may indeed be caused by enhanced activation of $H_2$ over Ni atoms. This promotes the reduction of $Ga_xO$ species (Fig. 3, reaction 5) and in consequence, shifts the kinetically controlled $Ga^0-Ga_xO$ redox process to the metallic side allowing for enhanced DRM over atomically dispersed Ni atoms in a matrix of Ga atoms. To further investigate this kinetically controlled redox process, the present $Ga_{12}Ni/SiC$ SCALMS was also tested with a 2:1 $CH_4:CO_2$ inlet ratio to compare the performance during DRM under a more reducing environment (Fig. S15). As expected, the equilibrated conversion of $CH_4$ and $CO_2$ lies above the one of DRM with equimolar feed as the concentration of $H_2$ form is increased.

The generally increased conversion levels may be explained by the almost five times higher loading of Ni in the $Ga_{12}Ni/SiC$ SCALMS (Table 1). In order to quantify the amount of Ni available at the surface of both Ga-Ni SCALMS, ab initio molecular dynamics (AIMD) simulations were carried out using slab models (Fig. 6a) at 900 °C with Ga/Ni ratios of 45 and 12, respectively, equalling those in the experiments. The systems were proven to be in a liquid state at this temperature by computation of the mean square displacement of the atoms from their initial positions, which showed a linear behaviour. Furthermore, density profiles for $Ga_{12}Ni$ and $Ga_{45}Ni$ were calculated (Fig. 6b). Expectedly, a surface depletion of Ni can be observed for both cases. This is in line with previous calculations for Pd, Pt, and Rh

SCALMS[20,26,33]. However, owing to the high mobility of the liquid system at elevated temperatures, the Ni atoms can easily access the surface from time to time to act as single atoms catalytic centres. To quantify the amount of Ni available at the surface, the number of Ni atoms that are present in the first layer of the density profile was evaluated (see Fig. 6a for definition of the first layer). The calculations suggest that 2.5% and 0.7% of the surface atoms are Ni atoms in the case of $Ga_{12}Ni$ and $Ga_{45}Ni$, respectively. Hence, the availability of Ni atoms at the surface of $Ga_{12}Ni$ is 3.6 times higher than for $Ga_{45}Ni$, which also approximates the increase in the conversion of $CO_2$ (Fig. 5). The relative depletion of Ni atoms at the surface is nearly equal in both cases. The higher Ni surface concentration in $Ga_{12}Ni$ simply results from the higher total amount of Ni in $Ga_{12}Ni$. This higher surface concentration may explain the higher catalytic activity going along with better long-term stability due to a higher rate of $H_2$ activation, which ultimately leads to a less oxidised state of the Ga-Ni interface.

In line with the conversion of $CH_4$, a stable conversion of $CO_2$ over the $Ga_{12}Ni/SiC$ SCALMS was achieved after 15 h TOS (Fig. 5). Once again, the level of $CO_2$ conversion is 50% higher than the conversion of $CH_4$ (Fig. S8) suggesting additional conversion of $CO_2$ via the RWGS reaction and oxidation of liquid $Ga^0$ with subsequent reduction of $Ga_xO$ species by $H_2$ (Fig. 3, reaction 4 & 5). However, the ratio in the conversion of both reactants is constant over 100 h TOS (Fig. S15), while the ratio slowly decreases for the $Ga_{45}Ni/SiC$ SCALMS indicating less conversion of $CO_2$ via oxidation of $Ga^0$ due to a continuous shift of the kinetically controlled $Ga^0-Ga_xO$ redox process to the oxidic side. Contrary, the increased Ni content in the $Ga_{12}Ni/SiC$ SCALMS allows for a kinetically controlled redox process on the metallic side due to sufficient $H_2$ activation. The enhanced consumption of $CO_2$ and $H_2$ in said oxidation-reduction-cycles also results in a lower $H_2/CO$ ratio in the produced synthesis gas, which stabilises at 0.45 (Fig. 5c). In contrast, the ratio obtained over the $Ga_{45}Ni/SiC$ SCALMS was continuously increasing from 0.3 to >0.8 over 20 h TOS. This observation, once again, may be linked to the detrimental effect of low Ni concentrations on the stability of SCALMS during DRM causing steady deactivation. The kinetically controlled redox process is far on the oxidic side and not reached over $Ga_{45}Ni/SiC$ SCALMS within 24 h TOS.

**Post-run catalyst characterisation**. Characterisation of the post-run Ga-Ni SCALMS by means of XRD and comparison with the as-prepared samples suggests an increased fraction of $Ga_2O_3$ after catalysis (Fig. 7). The patterns of the fresh samples only feature diffractions for this metal oxide close to the detection limit. A passivation layer after preparation of SCALMS has been reported[26,28,29] and the performance of the SCALMS during DRM suggests in situ reduction within the first hours TOS (Fig. 5). Contrary, the spent samples feature pronounced diffractions of $Ga_2O_3$ suggesting oxidation of the Ga-rich alloy during DRM. Potential passivation of the (bi-)metallic phase upon exposure to ambient air at room temperature during unloading of the catalyst from the reactor can be expected to form less oxide than the exposure to 500 °C during calcination. In addition, the $Ga_2O_3$ diffractions are even more pronounced for the $Ga_{45}Ni/SiC$ catalyst when compared to $Ga_{12}Ni/SiC$, which supports the hypothesised link between kinetically controlled redox process and Ni concentration.

The Ga-Ni combination was shown to play an essential role as an oxygen shuttle between $CO_2$ and $CH_4$. In particular, the reducibility of the $Ga_xO$ species is crucial and triggered by $H_2$ activation over the active metal. Here, the Ga-Ni SCALMS (and to a minor extent the Co-Ga SCALMS) were found to allow for

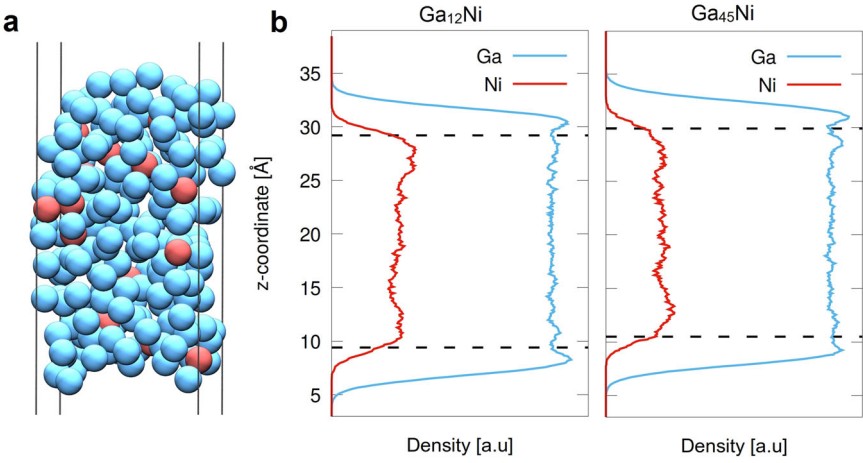

**Fig. 6 AIMD simulations of gallium-nickel supported catalytically active liquid metal solutions (SCALMS). a** Snapshot of the slab system with a unit cell of $Ga_{165}Ni_{15}$ corresponding to the experimental composition $Ga_{12}Ni$. Similarly, a $Ga_{176}Ni_4$ unit cell was used in the simulations of a $Ga_{45}Ni$ composition; Ga is shown in blue, Ni in red. **b** Density profiles for the compositions $Ga_{12}Ni$ (left) and $Ga_{45}Ni$ (right). The dashed lines mark the end of the first layer. The density profiles of Ni were scaled by a factor of 3 and 9 in the case of $Ga_{12}Ni$ and $Ga_{45}Ni$, respectively, for the sake of clarity.

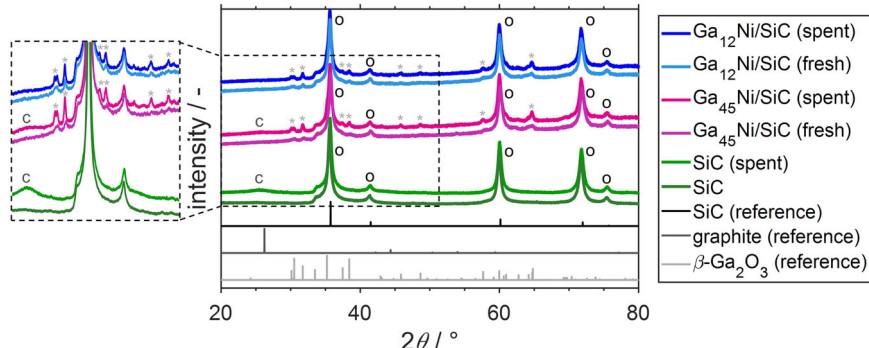

**Fig. 7 X-ray diffraction of fresh and spent catalysts.** X-ray diffractograms (Cu K-alpha radiation with $\lambda = 1.541$ Å) with an inset of the magnified intensity of Ga-Ni SCALMS employing a mesoporous SiC support, as well as the bare support material before and after application in dry reforming of methane together with reference patterns for SiC (o), graphite (C), and $\beta$-$Ga_2O_3$ (*).

closing the DRM catalytic cycle. A nearly constant conversion ratio of $CO_2/CH_4$ of approx. 1.5 was observed (Fig. S8) along with a constant hydrogen yield (Fig. S11). On the contrary, the Ga-Cu and Ga-Fe SCALMS underwent rapid deactivation resulting in a decreased conversion of $CO_2$ when compared to $CH_4$. Further, a steadily increasing $H_2$ yield indicates that these systems (after full oxidation) rather cause $CH_4$ pyrolysis under the applied conditions, which is expected to result in excessive coking. The increase in $H_2$ yield is also due to a lower WGS activity after approx. 10 h TOS, as the conversion of $CH_4$ remains constant (Fig. 2a).

TGA of the spent catalysts was performed in order to quantify the deposition of carbon during DRM at 900 °C for 24 h. The smallest weight loss during TPO in 21% $O_2/N_2$ was observed for the Ga-Ni SCALMS catalysts, while the catalysts with zero or low level of activity, as well as the SiC reference sample, displayed a high weight loss in the range of 16-19 wt.% (Fig. 8). In fact, TGA suggests a minor coke content of 0.7 wt.% for the $Ga_{12}Ni/SiC$, which may indicate a significant suppression of coking during DRM over SCALMS when compared to catalysts in the literature[17,54,55]. This is also evidenced during TGA of the spent $Ga_{12}Ni/SiC$ SCALMS after DRM at 900 °C for 100 h TOS, which exhibited a remarkably low coke content of 2.9 wt.% (Fig. S16). This superior coke resistance underlines the stable performance of this Ga-Ni SCALMS during DRM at 900 °C. In general, the coke content in the SCALMS systems correlates with the observed

activity during DRM suggesting the formation of highly reactive carbon species during DRM over Ni atoms. This process seems to be less efficient for Co and Cu dissolved in the liquid Ga matrix. Such highly active species may react with $CO_2$ via the Boudouard reaction (Fig. 3, reaction 2) or even initiate the reduction of oxidised $Ga_xO$ species (Fig. 3, reaction 6). Another reason may be the observed RWGS activity resulting in the formation of the oxidant $H_2O$, which may oxidise the liquid metal and carbon deposits alike. A similar suppression of coke formation over SCALMS has already been reported for propane dehydrogenation at lower temperatures of 450–550 °C[26,28,29]. The increased coke content of 12.4 wt.% in the $Ga_{45}Ni/SiC$ is most likely due to the detrimental transformation of the metallic Ga to oxidic species as the kinetically controlled redox process lies on the oxidic side and seemingly does not equilibrate over the studied TOS. In other words, coking may be reasonably suppressed by a liquid Ga-rich alloy, while its (partially) oxidised counterpart cannot restrict coking. This dependency was also observed by means of XRD (Fig. 7, inset). The typical (002) diffraction of carbon was observed for $Ga_{45}Ni/SiC$ and all less active catalysts (Fig. S17), but absent in the case of $Ga_{12}Ni/SiC$. The final weight gain of all samples during TGA (see range above 800 °C in Fig. 8) is a result of the enhanced $SiO_2$ passivation layer of the SiC support material at elevated temperatures[40,56,57].

The spent catalysts were also characterised by means of Raman spectroscopy (green laser with a wavelength of 532 nm) to obtain

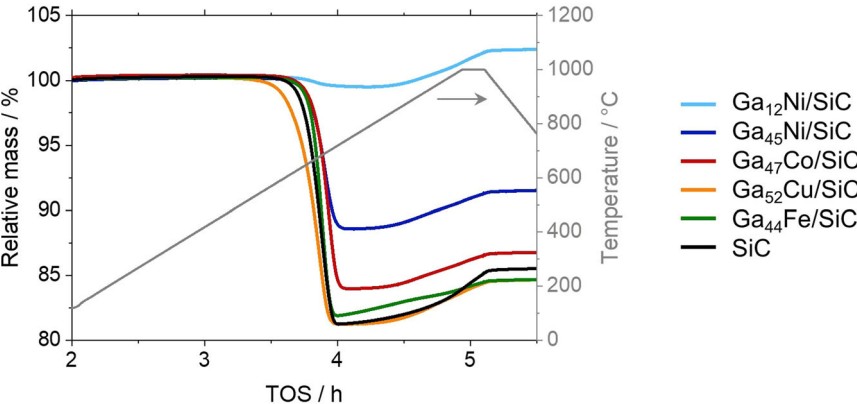

**Fig. 8 Thermogravimetric analysis via temperature-programmed oxidation.** Characterisation of spent SCALMS with various active metals employing a mesoporous SiC support, as well as the bare support material after application in dry reforming of methane.

additional information on coke deposits (Fig. S17). The bare SiC support material also features the characteristic D band (~1350 cm⁻¹) and the G band (~1600 cm⁻¹) of carbon. However, the intensity ratio of D over G band changes when comparing the spectra before and after catalytic application during DRM. The ratio lies in the range of 1.00–1.10 for the bare SiC, as well as the as-prepared SCALMS, and increases to 1.15–1.45 due to the coking of the catalysts (Table S1). Furthermore, said change was more pronounced for less active catalysts, which is in line with the results from TGA. In fact, the ratio was the smallest for the Ga₁₂Ni/SiC SCALMS, which displayed the highest activity during DRM. In addition, the spectra of all spent SCALMS and the bare SiC support after DRM feature the G' band at ~2700 cm⁻¹ (Fig. S18). No difference in carbon deposits may be identified by means of XRD (Fig. S17).

Potential loss of gallium from the SCALMS catalyst during DRM at 900 °C was investigated by analysing the metal loadings after catalytic testing using ICP-AES (Table 1). Noteworthy, the identified broad range in the coke content (Fig. 8) with an expected link to the hygroscopic behaviour of the spent samples[28] lowers the accuracy of the analysis of the metal content after catalytic application. Nevertheless, a certain loss in gallium was identified for all catalysts, which apparently scales with the activity of the catalyst, i.e. the loss of approx. 18% Ga was found to be larger for the more active Ga₁₂Ni SCALMS than for a molar ratio of Ga/Ni of 45. Furthermore, a loss of approx. 30% was identified for the long-term experiment of Ga₁₂Ni after 100 h TOS. This observation renders physical evaporation of liquid gallium unlikely, which is supported by the low vapour pressure of Ga⁰ at 900 °C of 6.2 · 10⁻⁷ bar[58]. The dependency on the reaction atmosphere rather suggests a strong link to the hypothesised redox cycles. In particular, the formation of more volatile Ga₂O, a likely intermediate formed during oxidation to Ga₂O₃ with CO₂ or subsequent re-reduction, may result in loss of gallium from the catalyst bed at reaction temperatures. H₂-containing atmospheres may even increase the volatility of gallium oxide[59]. The potential formation of volatile gallium hydrides cannot be excluded but has been reported for temperatures exceeding the herein applied conditions[60]. Our thermodynamic calculations also indicate a low feasibility of the formation of GaHₓ (Fig. S19). Hence, the redox cycles may continuously result in slow leaching from the catalyst bed. However, this process does not heavily affect the catalytic performance in the studied time range. Contrary, a minor increase in conversion between 60–100 h TOS (Fig. S15) was observed during the long-term testing and may indicate a slow and steady decrease of the Ga/Ni ratio below 12 due to the

preferential evaporation of Ga species. Such a change in the SCALMS composition may shift the kinetically controlled redox process further to the metallic side, which results in increased activity of the catalyst. Hence, the optimum Ga/Ni ratio for a balanced kinetically controlled redox process during DRM over SCALMS is expected to be below 8.4, which is the final composition of the catalyst after 100 h TOS (Table 1). The liquidus line of Ga-Ni at 900 °C is expected to have a Ga/Ni ratio of 3.1[39], which leaves significant space for further improvements of the SCALMS composition while complying with the fundamental idea of fully liquid-supported alloys. Furthermore, we assume that the tendency for Ga₂O evaporation reduces with higher Ni content of the alloy. Nevertheless, an additional experiment with a bimetallic catalyst with a molecular ratio of Ga/Ni of 0.74 was performed (Fig. S20). While the catalyst initially outperformed the Ga₄₅Ni/SiC SCALMS, rapid deactivation within 5 h TOS and excessive coke formation (16.2 wt.%; Fig. S21) were observed exhibiting the coking affinity of solid-based catalysts. Furthermore, and in line with our expectations, no loss in Ga during DRM was detected when comparing the loading before and after catalysis by means of ICP-AES (Table S2).

Finally, the Ga-Ni SCALMS materials were analysed by means of SEM-EDX after application in DRM to study morphological changes and evaluate the metal distributions. Note, that artefacts from solidification of the liquid alloy during cool-down and subsequent exposure to ambient air may affect this ex situ analysis. Nevertheless, the dispersion of gallium over the SiC support appears to be generally unchanged. However, the spent Ga₁₂Ni/SiC SCALMS features needle-type structures of several micrometres in length and diameters below 100 nm (Fig. 9; Fig. S22). These structures cannot be identified in the as-prepared samples nor in the Ga₄₅Ni/SiC SCALMS after 24 h TOS (Fig. 1; Figs. S4 & S23). According to elemental mapping via SEM-EDX, the needles consist of gallium and oxygen (Fig. 9). No nickel, carbon, or silicon can be detected suggesting the formation of GaₓO needles during DRM, most likely as Ga₂O₃ phase (Fig. S24). In addition, the Ga₄₅Ni/SiC SCALMS features some needles after DRM for an extended duration of 100 h (Fig. S25 & S26), probably because a longer run time was needed to locate the needles in post-run analysis. Hence, the formation of these structures depends on the extent and number of redox cycles as the Ga₁₂Ni SCALMS heavily oxidises and re-reduces due to the increased concentration of Ni in the Ga-rich supported alloys. As extended Ga leaching has been observed for both samples, a needle growth via the gas phase and volatile GaₓO species is highly likely. The volatility of oxidised gallium species has already

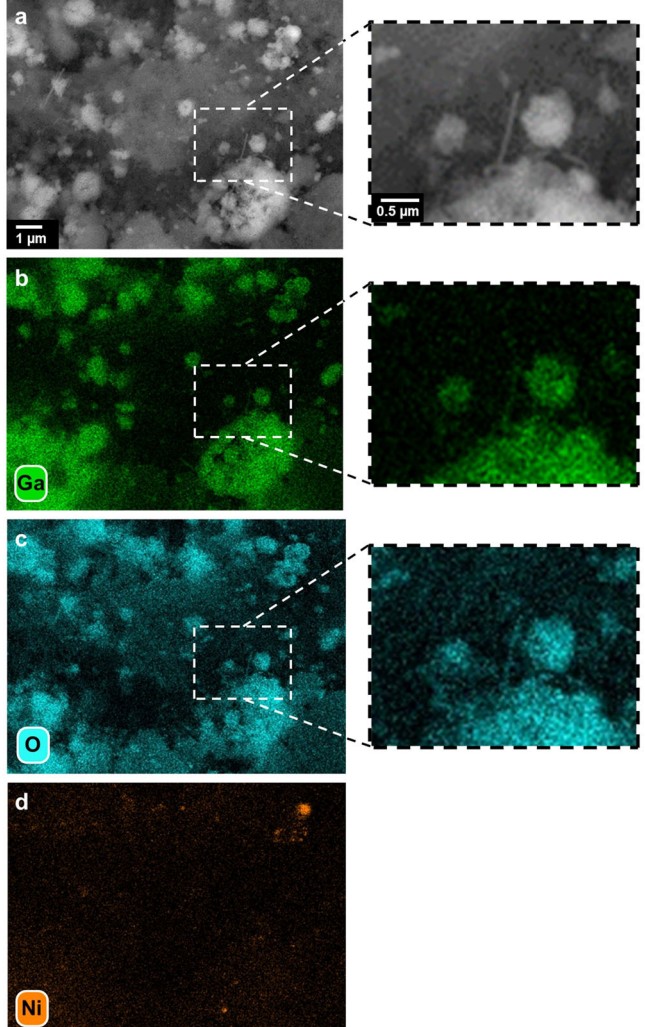

**Fig. 9 Morphology of spent gallium-nickel supported catalytically active liquid metal solutions (SCALMS). a** Backscattered scanning electron micrograph and elemental mapping of **b** gallium, **c** oxygen, and **d** nickel via energy-dispersive X-ray spectroscopy of $Ga_{12}Ni/SiC$ with metal loadings of Ga and Ni of 5.44 and 0.38 wt.%, respectively, after catalytic application in dry reforming of methane. Magnified insets show needle-type structures of gallium oxide.

been linked to the extent of redox cycles. Further, the preparation of β-$Ga_2O_3$ via chemical vapour deposition (CVD) using Ga and $H_2O$ has been reported in literature[61], while the presence of β-$Ga_2O_3$ in the spent SCALMS has been identified by means of XRD (Fig. 7). Other studies also describe the formation of needle-type structures of gallium oxide with comparable dimensions[62,63]. Lastly, the formation of needle- or platelet-type structures during water-induced deactivation of metal-based catalysts has been reported for other applications as well[64,65].

Aside from the needle-type structures, EDX mapping suggests a general co-location of nickel with the Ga-rich areas (Fig. 9), i.e., pronounced segregation during DRM is not observed. In contrast to the as-prepared SCALMS with randomly shaped Ni enrichments after impregnation and calcination (Fig. 4), the distribution of Ni after DRM suggests the formation of a liquid alloy under reaction conditions (Fig. 9). Such supported liquid alloy droplets can be expected to solidify during cool-down forming the herein observed spherical bimetallic phases. However, several Ni-enriched Ga-Ni structures may be observed indicating a certain

degree of redistribution of the bimetallic phase, which as well may also be an artefact from cool-down.

## Summary and conclusion

Supported Ga-rich liquid Ga-Co, Ga-Cu, Ga-Fe, and Ga-Ni alloys were prepared, characterised, and applied in DRM to assess the suitability of SCALMS for high-temperature applications and to push this supported liquid phase catalysis concept to boundaries. While physical evaporation of the liquid metal phase could be excluded, a certain degree of chemical leaching, most likely via oxidic gallium species, was observed. In general, all SCALMS were shown to oxidise during the applied DRM conditions. Only the supported Ga-Ni and, to a smaller extent, the Ga-Co alloy retained activity after a run-in phase. Our study indicates that this is due to the capability of the supported liquid Ga-Ni alloy to efficiently activate $H_2$ (formed during DRM) for the re-reduction of oxidised gallium species. For Ga-Ni SCALMS, the resulting kinetically controlled redox process could be shifted to the metallic side by increasing the concentration of Ni in the Ga-rich alloy, which promotes the activation of $H_2$ and enables stable conversion levels during DRM. A long-term experiment with $Ga_{12}Ni/SiC$ for 100 h time on stream demonstrated the general suitability of this SCALMS system for DRM, but the leaching of gallium species (in combination with the deposition of β-$Ga_2O_3$ needles) will affect the stability of Ga-Ni SCALMS with the here-studied composition in even longer operation. Further, the estimated intrinsic activity of Ni is inferior when compared to the literature.

Several important findings for future applications of SCALMS systems were made in this conceptual work:

– The SCALMS concept allows for the employment of less noble metals, such as Co and Ni, as the secondary active metal dissolved in Ga-rich liquid alloy phase. Ab initio molecular dynamics simulations showed a surface depletion of Ni accompanied by the dynamic temporary appearance of single Ni atoms at the surface, in analogy to the situation of SCALMS containing noble metals.

– Ga-based SCALMS are generally suitable for high-temperature operation at 900 °C, but volatile Ga-oxide species can cause Ga losses due to evaporation under an oxidising atmosphere.

– A reduced amount of coke formation was observed for Ga-Ni SCALMS when compared to the other catalysts with low or zero levels of activity. In fact, the $Ga_{12}Ni/SiC$ SCALMS with a high concentration of Ni in the supported liquid alloy almost completely suppressed coking during DRM due to the high dynamics of its supported liquid alloy interface.

## Methodology

**Materials**. Gallium pellets (≥99.9999% purity) were purchased from Alfa Aesar, cobalt(II) chloride ($CoCl_2$; ≥98.0% purity), copper(II) chloride ($CuCl_2$; ≥99.995% purity), iron(II) chloride ($FeCl_2$; ≥98.0% purity), and nickel(II) chloride ethylene glycol dimethyl ether ($C_4H_{10}Cl_2NiO_2$; ≥98.0% purity) were supplied from Sigma Aldrich, and isopropanol (≥99.3% purity) was purchased from Jäkle Chemie (Germany). All chemicals were used as received. Mesoporous 2 mm pellets of β-SiC (SIC3-E3-M) were purchased from SICAT (France), ground, and sieved to yield a particle size range of 500-630 μm.

**Preparation of catalysts**. Ga-based SCALMS materials were synthesised in a two-step procedure. The first part comprises the deposition of liquid Ga on the SiC support material. Ga nuggets of ~1.00 g were melted and dispersed in 100 mL isopropanol via

ultrasonication with a BRANSON-450D sonifier at 80% intensity (maximum power output: 450 W) for 10 min at 40 °C forming an emulsion. Subsequently, a defined amount of SiC (~13.00 g) was added to yield a targeted Ga loading. The mixture was stirred thoroughly for 1 h. Finally, the Ga-decorated SiC was dried with a rotary evaporator at 50 °C and 150 mbar for 2 h and calcined overnight at 500 °C[40]. The second part of the preparation procedure comprises the addition of the active metal via wetness impregnation of the Ga/SiC material. The corresponding amounts of metal precursors, namely metal chlorides (Co, Cu, Fe) and nickel(II) chloride ethylene glycol dimethyl ether (Ni), were dissolved in distilled water and isopropanol in a proportion of 1:10, added to the previously calcined Ga/SiC sample targeting a molar Ga/secondary metal ratio of 50 and mixed for 1 h. Subsequently, the solvents were evaporated in a rotary evaporator at 50 °C and 150 mbar for 2 h and the obtained SCALMS materials were calcined overnight at 500 °C.

**Characterisation of fresh and spent catalysts.** The metal loadings of prepared samples and post-run catalysts were determined by means of inductively coupled plasma atomic emission spectroscopy (ICP-AES) using a Ciros CCD (Spectro Analytical Instruments GmbH). The solid samples were digested with concentrated HCl:HNO:HF in a 3:1:1 volumetric ratio using microwave heating up to 220 °C for 20–40 min. The instrument was calibrated with standard solutions of Ni, Co, Fe, Cu, Ga prior to the measurements.

The BET surface area and BJH pore volume of the SiC support material were analysed by $N_2$ physisorption in a QUADROSORB SI Surface Area and Pore Size analyser (Quantachrome Instruments) with a degassing temperature of 200 °C.

Scanning electron microscopy (SEM) was carried out using a Phenom Desktop SEM (BSD detector, 15 kV voltage). The samples were deposited directly onto a conductive sticky carbon pad. In addition, scanning electron microscopy with elemental mapping via energy-dispersive X-ray spectroscopy (SEM-EDX) was performed on a ZEISS Cross Beam 540 Gemini II scanning electron microscope equipped with an EDX detector from Oxford Instruments Group. Silver paste (Acheson Silver DAG 1415) was used to enhance the electrical conductivity. Additionally, the samples were PVD carbon coated prior to the investigations. SEM images were obtained at 3 kV and 2 nA. For images acquired with a backscatter electron detector (BSD) a voltage of 20 kV and a current of 2 nA was used. EDX analysis was performed at 20 kV and 2 nA.

X-ray diffraction (XRD) was conducted using an X'Pert PRO (Philipps) equipped with a Cu anode ($\lambda_{K\alpha 1}$ = 1.54056 Å). The samples were placed in a sample holder and analysed in a continuous scan mode in the $2\theta$ range of 1.992° to 80.000° with a step size of 0.0167113° and a scan time of 1.11 s per step. XRD patterns were compared to references from the Crystallography Open Database (COD)[66], namely graphite (COD ID 1200017)[67], β-gallium(III) oxide (COD ID 2004987)[68], and β-silicon carbide (COD ID 1010995)[69].

Raman spectroscopy was conducted using an AvaRaman-PRB-532 (Avantes) probe with an AvaRaman-532HERO-EVO (Avantes) system. The Raman solution consists of a 532 nm (green) solid-state laser (Cobolt) and an AvaSpec-HERO (Avantes) spectrometer with a grating set of 1200 lines $mm^{-1}$ (HSC1200-0.75). The spectrometer is equipped with a 50 μm slit and the detected wavelength range is 534-696 nm. If not otherwise stated, Raman spectra were collected in 10 repetitions at 15 mW laser power with an exposure time of 20 s. Three different spots of the sample were analysed and averaged. Typically, all three spectra of the samples coincided.

Spent samples were analysed via temperature-programmed oxidation (TPO) in 21% $O_2/N_2$ in order to quantify carbon deposits formed during DRM. The weight change was monitored by means of thermogravimetric analysis (TGA) using a SETSYS Evolution high-performance modular TGA (Setaram). A total of 30–45 mg of the particular sample were placed in a quartz crucible, heated to 120 °C (10 °C $min^{-1}$) for 100 min to remove adsorbed $H_2O$. Subsequently, the temperature was increased with a heating rate of 5 °C $min^{-1}$ to 1000 °C for 10 min. The overall flow rate was 50 mL $min^{-1}$ throughout TPO.

**Catalytic testing.** Catalytic experiments were performed in a high-temperature set-up consisting of a tubular split furnace (Carbolite) with three heating zones of 20 cm length each. Quartz tubes (L: 100 cm, OD: 1.2 cm, ID: 1.0 cm) were used as reactors, while three pins at a height of 55.5 cm from the bottom end supported the catalyst bed. A total of 1.0 g of catalyst was placed in between two plugs of 0.1 g quartz wool. DRM was performed at 900 °C and 1 bar for 20 hours with an inlet gas composition of 20% $CH_4$ (≥99.9995% purity) and 20% $CO_2$ (≥99.9995% purity) in Ar (≥99.998% purity and further purified by an Agilent CP17974 gas clean filter) and an overall flow rate of 50 $mL_N$ $min^{-1}$ resulting in a gas hourly space velocity (GHSV) of 3000 $mL_N$ $h^{-1}$ $g_{cat}^{-1}$. The off-gas was quantitatively analysed in a micro gas chromatograph (I-GRAPHX PR; Industrial Graph Xolutions, Germany) using Ar as the internal standard.

**Computational methods.** Density-functional theory (DFT) based ab initio molecular dynamics (AIMD) simulations were performed employing the Vienna Ab Initio Simulation Package (VASP) using the projector augmented wave (PAW) method to represent the ion cores and a plane wave basis set with a kinetic energy cutoff of 300 eV[70–72]. The exchange-correlation function of Perdew, Burke and Ernzerhof (PBE) was employed[73]. The GaNi systems were simulated using periodic slab models with a tetragonal unit cell ($12.69 \times 12.69 \times 40$ Å) comprising 180 Ga atoms with 15 and 4 Ni atoms in case of $Ga_{12}Ni$ and $Ga_{45}Ni$, respectively. A vacuum layer of 12 Å was added in the direction perpendicular to the surface to ensure that periodic images do not interact. To integrate the equations of motion a Verlet algorithm was used with a time step of 5 fs. After sufficient equilibration, a Nosé-Hoover thermostat was applied in order to simulate a canonical ensemble at 900 °C[74]. To sample the first Brillouin zone a Γ-containing 2x2x1 k-point mesh was used. The SCF convergence criterion was set to $10^{-7}$ eV. A total of 10 independent trajectories were sampled for each composition, every trajectory running for at least 150 ps of production time and results were averaged over all trajectories.

### Data availability
All relevant data are available from the authors upon request. Please contact the corresponding author.

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

## Acknowledgements

Financial support by the European Research Council (Project 786475: Engineering of Supported Catalytically Active Liquid Metal Solutions) and the German Research Foundation (DFG) in the frame of the Collaborative Research Centre 1452 Catalysis at Liquid Interfaces (CLINT) is gratefully acknowledged. The authors thank Narayanan Raman and Susanne Pachaly for conducting the N₂ physisorption and XRD measurements, respectively.

## Author contributions

M.W.: conceptualisation; investigation; data curation; formal analysis; validation; visualisation; writing—original draft. A.L.d.O.: Investigation; data curation; formal analysis; writing—review & editing. N.T.: investigation; formal analysis; validation; writing—review & editing. S.M.: investigation; data curation; formal analysis; visualisation. M. Heller: Investigation; data curation; formal analysis; visualisation. SKA: investigation; data curation; formal analysis. AS: resources. PF: resources; supervision. AG: resources; supervision; writing—review & editing. M. Haumann: conceptualisation; project administration; resources; validation; writing—review & editing. PW: conceptualisation; funding acquisition; project administration; resources; validation; writing—review & editing.

## Funding

## Competing interests

The authors declare no competing interests.
