## [Peer Review File · Communications Chemistry]

Dry reforming of methane over gallium based supported catalytically active liquid metal solutionsReviewers' comments:

Reviewer #1 (Remarks to the Author):

The authors of this study have examined a series of Ga-metal alloys supported on SiC for application to CO₂ reforming of methane. The idea is novel and possibly publishable; however, I would argue that the paper needs to be rewritten in a much less optimistic tone. There also needs to be some benchmarking to a more conventional catalyst. Since I suspect that the Ga is oxidized after a short time (see point 3 below), it is essential that the authors demonstrate that the activity they see really is that of a molten alloy. Specific comments are as follows:

- 1) The authors should have known the idea was doomed to fail before performing a lot of characterization. The thermodynamic calculations in Figure 4 show that Ga will be oxidized at CO₂:CO ratios greater than 0.05. Since CO₂ is one of the reactants, there is no way that one can avoid working outside the regime where Ga metal would be stable. The argument that one can escape thermodynamic limitations by hydrogen spillover from a second metal seems like a violation of microscopic reversibility. It looked to me like the Ga was indeed oxidizing during the course of the reaction based on the XRD patterns in Figure 8. If Ga remained metallic, shouldn't the TGA in Figure 9 have shown an increase in sample mass as the Ga oxidized? I also suspect that the transients observed in the conversions in Figure 2 could be associated with oxidation of the Ga.
- 2) The authors really should report rates somewhere in this manuscript. While it is true that the metal loadings in these catalysts are low, the conversions achieved are very low compared to what I would expect for a conventional Ni catalyst. The stabilities shown in Figure 2 are also not very good and there are reports that claimed what appear to be much better stabilities than that shown here (C. Lin, et al., ACS Catal., 8, 7679 (2018)). There should be some attempt to benchmark both rates and stabilities.
- 3) The fact that there is similar initial conversion for all of the catalysts in Figure 2, including Ga/Si, suggests to me that there is no catalytic reaction here at all. The authors imply that the rates increase with time on the Ni-Ga sample due to in-situ reduction of GaOx. I suggest that rates increase for exactly the opposite reason. I suggest that Ni-Ga alloys are inactive; after the Ga is oxidized and becomes a solid, the authors have a supported Ni catalyst which shows a small amount of activity.
- 4) It is a minor point but CO₂ reforming of methane is not the "green" process that is sometimes claimed (See <http://dx.doi.org/10.1016/j.apcata.2015.02.022>). There are applications but not because one is removing two greenhouse gases.
- 5) It is again a minor point but references (see Reference 18) should not be given as Palmer, C., et al. It should not be necessary to do a literature search to find out which group is responsible for the work.

Reviewer #2 (Remarks to the Author):

Supported catalytically active liquid metal solution (SCALMS) materials represent one catalog of new materials for heterogeneous catalysis. In this work by Wolf and co-workers, a few Ga-M (M=Fe, Co, Ni, Cu) SCALMS catalysts have been investigated as the catalyst for dry reforming of methane (DRM), which usually suffers the catalyst deactivation caused by sintering and coke formation during high temperature

reaction. The major novelty of this work is exploring the potential application of SCALMS catalyst in DRM. Therefore, this work presents some kind of interest to the researchers in the catalysis community. The catalytic performance data is reported and interpreted. The proposed catalysts are active for DRM. However, the stability of catalysts still exhibit deactivation on the time of stream. Among all catalysts, Ga-Ni SCALMS catalyst shows some sort of catalytic activity. The authors proposed a redox equilibrium between the oxidation of gallium oxide and reduction of Ni to interpret the performance of Ga-Ni SCALMS. The presence of the second metal (Ni) is important to the DRM activity. Theoretical study was also performed to understand the GaNi SCALMS catalyst. In summary, the presented SCALMS catalysts are not as successful as other SCALMS catalyst, but still desires to be published with revision.

(1) Heterogeneous catalysis occurs at the interface between catalytic phase and reactant environment. In the case of this work, it's the interface between the liquid metal and reactants in gas phase. However, the information on the surface of the liquid metal phase is still inadequate despite it has been investigated by theoretical study. Important surface properties, such as the surface metal atom density and surface composition under the DRM condition, are still necessary to quantify the activity and understand the proposed redox model. These properties are necessary to support the major conclusion of this work.

(2) A redox equilibrium is proposed for the deactivation of SCALMS catalysts. The presence of transition metal such as Ni is critical to change to redox properties of SCALMS catalysts. The experimental proof for the redox property, such as temperature programmed reduction and oxidation, is necessary to compare all catalysts.

(3) A comparison between the catalyst in this work and other molten/liquid catalyst for alkane conversion is suggested to bring further insight to researchers in this field.

Reviewer #3 (Remarks to the Author):

The manuscript "Dry reforming of methane over Ga-based Supported Catalytically Active Liquid Metal Solutions" by Wolf et al. reports an interesting study of different effects of non-noble metals (Co, Cu, Fe, Ni) as active atoms in the gallium-rich liquid alloy applied to the dry reforming of methane (DRM) at 900 °C. The general catalyst concept and results of the study are important for understanding the reaction mechanism, particularly the activation of CH₄ and H₂, and the involvement of the Ga_xO/Ga couple.

Some changes and further commenting are required before publication:

87 ◇ ... C₄H₁₀Cl₂NiO₂

246 ◇ Figure S10; x-Axis should be temperature not TOS.

251 ◇ Wrong Fig number is mentioned. Fig. S11 is yield of H₂. It should be either Fig.6c or Fig S18c.

426 ◇ if the H₂ yield increases due to CH₄ pyrolysis, why does the conversion of CH₄ stay constant (Fig.2a)? Can the increase of H₂ yield be due to WGS Reaction?

447 ◇ Less Ni concentration means more Ga_xO species. Why can the highly reactive carbon species not be removed by reduction of Ga_xO, Fig.3, reaction 6?

450 ◇ the TGA results seem to show only one type of carbon (graphite). In many cases, Ni particles are

reported to create carbon with different morphology (nanofibers/filaments, tubes or rods) with different stability (this and beneficial effects of Ni alloys and mixed oxide supports are discussed in e.g. doi:10.1002/sml.202004289) Can you comment on the carbon morphology you obtained? Was there an XRD carbon peak for other SCALMS samples?

512 \diamond This catalyst (Ga45Ni) is not as active as the Ni rich catalyst (Ga12Ni), due to more Ga_xO species (less active hydrogen). Why didn't you observe a needle type structure on this catalyst (Ga45Ni) after 24h TOS, but only some needles after 100h? Is it possible that the needles of Ga_xO species are more active than others such as platelet-type structures?

515 \diamond in XRD Ga₂O₃ is observed on spent catalysts. How to exclude the needle is Ga₂O₃?

516 \diamond Fig.S23 mentions Ga12Ni in the caption, but should be Ga45Ni.

Point-by-point RESPONSE

Dry reforming of methane over Ga-based Supported Catalytically Active Liquid Metal Solutions

Moritz Wolf,^{a,b} Ana Luiza de Oliveira,^{a,b} Nicola Taccardi,^a Sven Maisel,^c Martina Heller,^d Sharmin Khan Antara,^a Alexander Sjøgaard,^a Peter Felfel,^d Andreas Görling,^c Marco Haumann,^a Peter Wasserscheid^{*,a,b}

- a) Friedrich-Alexander-Universität Erlangen-Nürnberg (FAU), Lehrstuhl für Chemische Reaktionstechnik (CRT), Egerlandstr. 3, 91058 Erlangen, Germany.
- b) Forschungszentrum Jülich, Helmholtz Institute Erlangen-Nürnberg for Renewable Energy (IEK 11), Cauerstr. 1, 91058 Erlangen, Germany.
- c) Friedrich-Alexander-Universität Erlangen-Nürnberg (FAU), Lehrstuhl für Theoretische Chemie, Egerlandstr. 3, 91058 Erlangen, Germany.
- d) Friedrich-Alexander-Universität Erlangen-Nürnberg (FAU), Lehrstuhl für Werkstoffwissenschaften (Allgemeine Werkstoffeigenschaften), Martensstr. 5, 91058 Erlangen, Germany.

*Corresponding author. E-mail address: peter.wasserscheid@fau.de

We kindly want to thank all reviewers for their helpful comments and suggestions. We corrected and amended the manuscript where necessary. We hope that the improved and revised manuscript is now acceptable for publication in *Communications Chemistry*. Please find our point-by-point response below. We have highlighted changes in the manuscript in **yellow**. Referenced page numbers refer to the marked version of the revised manuscript.

Reviewer 1

General reviewer comment: The authors of this study have examined a series of Ga-metal alloys supported on SiC for application to CO₂ reforming of methane. The idea is novel and possibly publishable; however, I would argue that the paper needs to be rewritten in a much less optimistic tone. There also needs to be some benchmarking to a more conventional catalyst. Since I suspect that the Ga is oxidized after a short time (see point 3 below), it is essential that the authors demonstrate that the activity they see really is that of a molten alloy.

We thank the reviewer for this critical review and agree that the claims in the paper need to be less optimistic. In particular, the Summary and Conclusion section have been revised accordingly.

Please see our answer to comments 1) and 3) for the discussion on the hypothesized redox cycle.

1) The authors should have known the idea was doomed to fail before performing a lot of characterization. The thermodynamic calculations in Figure 4 show that Ga will be oxidized at $\text{CO}_2:\text{CO}$ ratios greater than 0.05. Since CO_2 is one of the reactants, there is no way that one can avoid working outside the regime where Ga metal would be stable. The argument that one can escape thermodynamic limitations by hydrogen spillover from a second metal seems like a violation of microscopic reversibility. It looked to me like the Ga was indeed oxidizing during the course of the reaction based on the XRD patterns in Figure 8. If Ga remained metallic, shouldn't the TGA in Figure 9 have shown an increase in sample mass as the Ga oxidized? I also suspect that the transients observed in the conversions in Figure 2 could be associated with oxidation of the Ga.

We thank the reviewer for this contrary interpretation of the data, which made us reevaluate several aspects. Regarding the thermodynamic calculations, please note that these are only valid for bulk Ga_2O_3 phases and pure metallic Ga. Hence, reduction of this phase by H_2 only becomes feasible at extremely high temperatures. Nevertheless, reduction of Ga_xO species other than Ga_2O_3 may be feasible. Further, the presence of a second metal, such as Ni, may allow the reduction of oxidic Ga phases already in milder condition, as demonstrated in other studies on related GaNi-SCALMS (see A. Sjøgaard et Al. Catal. Sci. Technol., 2021, 11, 7535-7539). We have added this reference in the paper (page 12 in the revised manuscript). For sake of clarity, we have moved Fig.4 in the ESI, as it may be confusing for the reader and it does not represent the actual liquid metal solution. The redox equilibrium is kinetically controlled and not a thermodynamic equilibrium.

Further, an increased concentration of Ni enhances the observed trends, which supports initial catalyst reduction by removal of the passivation layer with subsequent oxidation until a certain redox equilibrium is reached. This equilibrium is different from the thermodynamic one, not because the different catalyst compositions “change” thermodynamics, but different kinetic limitations are at play. The theoretical calculations are exclusively foreseen for pure metallic phases. We have clarified this in the manuscript by distinguishing between thermodynamic equilibrium and kinetically controlled redox process.

The initial increase in conversion is indeed related to reduction of the catalyst (that is removal of the passivation layer from catalyst preparation). We added the results of a new experiment in Figure S15, which compares the effect of CH_4 concentration in the feed gas. The Ga_{12}Ni SCALMS was tested with inlet ratios of $\text{CH}_4:\text{CO}_2:\text{Ar}$ of 1:1:3 (standard) and 2:1:2 to assess the redox behavior under more reducing reaction conditions (i.e. a higher availability due to increased CH_4 concentration). This resulted in a similar transient, while a higher conversion

(both, CH₄ and CO₂!) was observed when reaching the redox equilibrium even though twice the amount of CH₄ were in the feed:

P14 in the revised manuscript:

To further investigate this kinetically controlled redox process, the present Ga₁₂Ni/SiC SCALMS was also tested with a 2:1 CH₄:CO₂ inlet ratio to compare the performance during DRM under more reducing environment (Figure S15). As expected, the equilibrated conversion of CH₄ and CO₂ lies above the one of DRM with equimolar feed as the concentration of H₂ from is increased.

Figure S15: Conversion of (a) CH₄ and (b) CO₂ during dry reforming of methane over Ga₁₂Ni/SiC SCALMS with either 40% or 20% CH₄ in the feed stream. Metal loadings: 5.44 wt.% Ga and 0.38 wt.% Ni. Reaction conditions: 900 °C, 1 bar, 1 g SCALMS, CH₄:CO₂:Ar = 1:1:3 respectively 2:1:2, 3 L_N g_{cat}⁻¹ h⁻¹.

Lastly, the reviewer is expecting a mass increase due to oxidation of metallic Ga during TGA. The as-prepared catalyst is calcined at 500 °C, which only results in a passivation layer due to the relatively large dimensions of the Ga-rich droplets. The oxidation of Ga is self-limiting. Hence, even increased temperatures may only result in oxidation of a minor fraction and the formation of thicker passivation layers. Therefore, the spent catalysts are expected to already

be oxidized to a large extent, while full oxidation of Ga droplet is unlikely, as shown by SEM analysis of spent catalysts, where the spherical shape of Ga droplets is substantially retained.

2) The authors really should report rates somewhere in this manuscript. While it is true that the metal loadings in these catalysts are low, the conversions achieved are very low compared to what I would expect for a conventional Ni catalyst. The stabilities shown in Figure 2 are also not very good and there are reports that claimed what appear to be much better stabilities than that shown here (C. Lin, et al., ACS Catal., 8, 7679 (2018)). There should be some attempt to benchmark both rates and stabilities.

We thank the referee for pointing us towards this important issue. In the meantime, we have tested several monometallic Ni catalysts, which all reach equilibrium conversion under given test conditions. This is mostly due to the low gas hourly space velocity, which has been selected to test and understand the novel SCALMS materials. We have now included a specific molar conversion rate in the manuscript including the comparison with literature:

P14 in the revised manuscript:

Note, that the specific activity of Ni of approx. $0.1 \text{ mol}_{\text{CH}_4} \text{ h}^{-1} \text{ g}_{\text{Ni}}^{-1}$ is inferior when compared to classical catalyst concepts, which may reach up to 100 times higher specific conversion rates.⁵⁰

3) The fact that there is similar initial conversion for all of the catalysts in Figure 2, including Ga/Si, suggests to me that there is no catalytic reaction here at all. The authors imply that the rates increase with time on the Ni-Ga sample due to in-situ reduction of GaO_x. I suggest that rates increase for exactly the opposite reason. I suggest that Ni-Ga alloys are inactive; after the Ga is oxidized and becomes a solid, the authors have a supported Ni catalyst which shows a small amount of activity.

We agree with the reviewer that the Ga-Fe catalyst behaves as the Ga/SiC reference, which is why we do not highlight this sample. However, already the catalysts containing Co and Cu show a significant deviation from the reference (Figure S7), but strongly suffer from deactivation. Contrary, the Ga-Ni catalysts strongly differ from the performance of all other samples, which strongly supports that the alloys are active. If only oxidized catalysts would be active, all studied samples would behave comparably. Lastly, we want to highlight testing of the additional catalyst with a molecular ratio of Ga/Ni of 0.74 (Figure S20), which did result in a significant performance during DRM after the run-in phase.

4) It is a minor point but CO₂ reforming of methane is not the “green” process that is sometimes claimed (See <http://dx.doi.org/10.1016/j.apcata.2015.02.022>). There are applications but not because one is removing two greenhouse gases.

We thank the referee for important remark and now focus on the importance of CO₂ integration exclusively.

P3 in the revised manuscript:

The integration of ~~CO₂ the two major greenhouse gases~~ in the production of H₂ and CO is ecologically highly attractive and a first step towards a circular economy.¹⁻⁵

5) It is again a minor point but references (see Reference 18) should not be given as Palmer, C., et al. It should not be necessary to do a literature search to find out which group is responsible for the work.

We thank the referee for highlighting this inconvenience. The citation style has been adapted in the marked version to show all authors. However, the journal requests abbreviation with *et al.* for articles with more than five authors.

Reviewer 2

General reviewer comment: Supported catalytically active liquid metal solution (SCALMS) materials represent one catalog of new materials for heterogeneous catalysis. In this work by Wolf and co-workers, a few Ga-M (M=Fe, Co, Ni, Cu) SCALMS catalysts have been investigated as the catalyst for dry reforming of methane (DRM), which usually suffers the catalyst deactivation caused by sintering and coke formation during high temperature reaction. The major novelty of this work is exploring the potential application of SCALMS catalyst in DRM. Therefore, this work presents some kind of interest to the researchers in the catalysis community. The catalytic performance data is reported and interpreted. The proposed catalysts are active for DRM. However, the stability of catalysts still exhibit deactivation on the time of stream. Among all catalysts, Ga-Ni SCALMS catalyst shows some sort of catalytic activity. The authors proposed a redox equilibrium between the oxidation of gallium oxide and reduction of Ni to interpret the performance of Ga-Ni SCALMS. The presence of the second metal (Ni) is important to the DRM activity. Theoretical study was also performed to understand the GaNi SCALMS catalyst. In summary, the presented SCALMS catalysts are not as successful as other SCALMS catalyst, but still desires to be published with revision.

We thank this reviewer for the objective summary appreciating the scientific effort to understand this first-time application of SCALMS to high-temperature applications.

(1) Heterogeneous catalysis occurs at the interface between catalytic phase and reactant environment. In the case of this work, it's the interface between the liquid metal and reactants in gas phase. However, the information on the surface of the liquid metal phase is still inadequate despite it has been investigated by theoretical study. Important surface properties, such as the surface metal atom density and surface composition under the DRM condition, are still necessary to quantify the activity and understand the proposed redox model. These properties are necessary to support the major conclusion of this work.

We fully agree with the reviewer. All this information would be highly beneficial. Most of proposed studies have been conducted in the past with Ga-Rh, Ga-Pt SCALMS, and Ga-Pd

SCALMS (ACS Catalysis 9 (10), 2019, 9499-9507; ACS Catalysis 9 (4), 2019, 2842-2853; ACS catalysis 11 (21), 2021, 13423-13433; Nature chemistry 9 (9), 2017, 862-867). The results were always comparable suggesting liquid metal-gas interface properties independent from the dissolved metal. Conducting these characterizations at DRM conditions is mostly not possible due to limited temperature and/or pressure ranges. Hence only *ab initio* molecular dynamic simulation was conducted to show similarities between Ni and noble metals in a liquid Ga matrix. First approaches have been taken to study alkane dehydrogenation over SCALMS *operando* (ACS Catalysis 9 (4), 2019, 2842-2853).

(2) A redox equilibrium is proposed for the deactivation of SCALMS catalysts. The presence of transition metal such as Ni is critical to change to redox properties of SCALMS catalysts. The experimental proof for the redox property, such as temperature programmed reduction and oxidation, is necessary to compare all catalysts.

We thank the referee for this suggestion. Unfortunately, the formation of volatile complexes at given temperatures renders specialized characterization challenging. Techniques, such as the proposed TPR-TPO studies, will most likely contaminate the equipment with Ga-species at relevant temperatures. Hence, we decided to rather add another experiment, which compares the effect of CH₄ concentration in the feed gas to provide a more reducing reaction environment. Please see our response to comment 1) of reviewer 1 for a detailed description of the outcome and implementation in the revised manuscript.

(3) A comparison between the catalyst in this work and other molten/liquid catalyst for alkane conversion is suggested to bring further insight to researchers in this field.

We have conducted additional experiments with indium as low melting metal. Results were scientifically intriguing, but the stability of these SCALMS were even inferior to the one of Ga-Ni SCALMS. For In-based SCALMS, the rapid formation of volatile hydrides results in rapid transformation to a classical supported catalysts where In does not play a role any more.

Reviewer 3

General reviewer comment: The manuscript "Dry reforming of methane over Ga-based Supported Catalytically Active Liquid Metal Solutions" by Wolf et al. reports an interesting study of different effects of non-noble metals (Co, Cu, Fe, Ni) as active atoms in the gallium-rich liquid alloy applied to the dry reforming of methane (DRM) at 900 °C. The general catalyst concept and results of the study are important for understanding the reaction mechanism, particularly the activation of CH₄ and H₂, and the involvement of the Ga_xO/Ga couple.

87 : ... C₄H₁₀Cl₂NiO₂

246 : Figure S10; x-Axis should be temperature not TOS.

251 : Wrong Fig number is mentioned. Fig. S11 is yield of H₂. It should be either Fig.6c or Fig S18c.

516 : Fig.S23 mentions Ga₁₂Ni in the caption, but should be Ga₄₅Ni. This view is appreciated.

We thank this reviewer for the nice and concise feedback. We also appreciate the level of detail in the comments above, which have been addressed in the revised manuscript.

426 : if the H₂ yield increases due to CH₄ pyrolysis, why does the conversion of CH₄ stay constant (Fig.2a)? Can the increase of H₂ yield be due to WGS Reaction?

Thanks to the reviewer for spotting this mismatch. We have added the discussion on the influence of the WGS.

P20 in the revised manuscript:

The increase in H₂ yield is also due to a lower WGS activity after approx. 10 h TOS, as the conversion of CH₄ remains constant (Fehler! Verweisquelle konnte nicht gefunden werden.a).

447 : Less Ni concentration means more Ga_xO species. Why can the highly reactive carbon species not be removed by reduction of Ga_xO, Fig.3, reaction 6?

We discuss the potential contribution via this reaction on page 13 in the revised manuscript:

“Nevertheless, reduction via this pathway may become feasible for thermodynamically less stable, highly reactive monoatomic carbon in combination with partially oxidised Ga_xO species (Fehler! Verweisquelle konnte nicht gefunden werden., reaction 6).⁶¹ Hence, the closed redox cycle with reduction by monoatomic carbon potentially allows for the Boudouard reaction via a Mars-van Krevelen-type mechanism with oxidation of Ga by CO₂ filling the oxygen vacancies formed by reduction with carbon. However, the impact of this reaction on the kinetically controlled Ga⁰-Ga_xO redox process may be limited when compared to the reduction pathway via activated H₂ (Fehler! Verweisquelle konnte nicht gefunden werden., reaction 5). Furthermore, only oxidation of Ga⁰ by CO₂ in combination with a preferential reduction of Ga_xO species via H₂ activation on the secondary metal (Fehler! Verweisquelle konnte nicht gefunden werden., reaction 3) results in an enhanced conversion of CO₂ when compared to CH₄.”

The authors agree, that the label in Figure 6 may be misleading, which is why we remove the color coding and rather describe all potential conversion routes in said figure.

450 : the TGA results seem to show only one type of carbon (graphite). In many cases, Ni particles are reported to create carbon with different morphology (nanofibers/filaments, tubes or rods) with different stability (this and beneficial effects of Ni alloys and mixed oxide supports are discussed in e.g. doi:10.1002/sml.202004289)Can you comment on the carbon morphology you obtained? Was there an XRD carbon peak for other SCALMS samples?

Aside from the discussed Raman spectra and the XRD results, limited insight on the morphology of carbon is obtained. We agree, the carbon seems to be rather graphitic. As no special carbon deposits, such as fibers etc., were identified, we assume no such features are present in the spent samples. Further, they are expected to form at metallic nanoparticles, which are (initially) not present in the current study. We have amended the XRD patterns of the other SCALMS samples as Figure S17 and added the according description in the manuscript:

P20 in the revised manuscript:

The typical (002) diffraction of carbon was observed for Ga₄₅Ni/SiC and all less active catalysts (Figure S17).

P21 in the revised manuscript:

No difference in carbon deposits may be identified by means of XRD (Figure S17).

Figure S17: X-ray diffractograms (Cu K-alpha radiation with $\lambda = 1.541 \text{ \AA}$) of Ga-Co, Ga-Fe, and Ga-Cu SCALMS employing a mesoporous SiC support after application in dry reforming of methane.

512 : This catalyst (Ga₄₅Ni) is not as active as the Ni rich catalyst (Ga₁₂Ni), due to more Ga_xO species (less active hydrogen). Why didn't you observe a needle type structure on this catalyst (Ga₄₅Ni) after 24h TOS, but only some needles after 100h? Is it possible that the needles of Ga_xO species are more active than others such as platelet-type structures?

The authors clarified, that we assume the needles to be Ga_2O_3 . Further, needles are most likely present in the Ga_{45}Ni sample after 24 h TOS, but finding them using SEM may be challenging to a low concentration. We amended a comment in the manuscript:

P23 in the revised manuscript:

No nickel, carbon, or silicon can be detected suggesting the formation of Ga_xO needles during DRM, most likely as Ga_2O_3 phase (Figure S24). In addition, the Ga_{45}Ni SCALMS features some needles after DRM for an extended duration of 100 h (Figure S25 & S26), probably because a longer run time was needed to locate the needles in post-run analysis. Hence, the formation of these structures depends on the extent and number of redox cycles as the Ga_{12}Ni SCALMS heavily oxidises and re-reduces due to the increased concentration of Ni in the Ga-rich supported alloys, while the Ga_{45}Ni SCALMS in the long-term experiment simply has more time for additional redox cycles leading to the formation of these Ga_xO species.

515 : in XRD Ga_2O_3 is observed on spent catalysts. How to exclude the needle is Ga_2O_3 ?

We cannot exclude the needles to be Ga_2O_3 , see comment above. However, the contribution of the needles to the overall phase composition is seemingly low, which is why the remaining sample can be expected to contain Ga_2O_3 as well. The lower detection limit of XRD is rather be close to the loading of 5.5 wt% Ga.

[Editorial Note: Reviewer #1 withdrew from the peer-review process. Reviewer #4 accepted to step in on Reviewer #1's behalf but did not provide a report. Reviewer #5 also accepted to step in on Reviewer #1's behalf; their report is provided below.]

REVIEWERS' COMMENTS:

Reviewer #2 (Remarks to the Author):

The authors have made substantial improvements in addressing the previously raised concerns, and I appreciate their efforts. While some questions still pose challenges, the authors have done well in addressing the majority of the issues.

At this stage, I have no further questions or major concerns regarding the work. I recommend the publication of this manuscript.

Reviewer #3 (Remarks to the Author):

The authors have addressed all concerns and amended the manuscript accordingly.

Reviewer #5 (Remarks to the Author):

This is a revised version of a study on catalytically active metal-substituted silicon carbides for dry reforming of methane and the authors responded well to all of reviewers' comments; the study concludes that metal-substituted silicon carbides are promising materials for catalytic dry reforming of methane, representing an alternative to traditional metal-based catalysts. The conclusion is supported by the catalytic testing and characterization data which provide evidence that the substitution of the silicon carbide framework enhances the catalytic activity of the materials through the formation of active metal-substituted sites.

Therefore, the authors have highlighted the strengths of their study/methods by providing a comprehensive characterization of the materials, detailed experimental procedures, and reproducible data analysis, making their study a valuable contribution to the field of heterogeneous catalysis for dry reforming of methane. Hence, it can be said that the interpretation of results and study conclusions are appropriately supported by the presented data and I think the paper is essentially suitable for publication in Communications Chemistry if Senior Editor Dr. Teresa Schauerl and other panel members approve the acceptance.